

# ChinaCropPhen1km: A high-resolution crop phenological dataset for three staple crops in China during 2000-2015 based on LAI products

Yuchuan Luo [1], Zhao Zhang [1*], Yi Chen [2], Ziyue Li [1], Fulu Tao [2, 3, *]

[1]State Key Laboratory of Earth Surface Processes and Resource Ecology, Key Laboratory of Environmental Change and Natural Hazards, Faculty of Geographical Science, Beijing Normal University, Beijing 100875, China
[2]Key Laboratory of Land Surface Pattern and Simulation, Institute of Geographical Sciences and Natural Resources Research, Chinese Academy of Sciences, Beijing 100101, China
[3]College of Resources and Environment, University of Chinese Academy of Sciences, Beijing 100049, China

*Correspondence to*: Zhao Zhang (sunny_zhang@bnu.edu.cn)

**Abstract.** Crop phenology provides essential information for land surface phenology dynamics monitoring and modelling, and crop management and production. Most previous studies mainly investigated crop phenology at site scale, however, land surface phenology dynamics monitoring and modelling at a large-scale need a high-resolution spatially explicit information on crop phenology dynamics. In this study, we proposed a method to retrieve 1km-grid crop phenological dataset for three main crops from 2000 to 2015 based on GLASS LAI products. First, we compared three common smoothing methods and chose the most suitable methods for different crops and regions. Then, we developed an optimal filter-based phenology detection (OFP) approach which combined both inflexion- and threshold-based method and detected the key phenological stages of three staple crops at 1km spatial resolution across China. Finally, we established a high resolution gridded-phenology product for three staple crops in China during 2000-2015, named as ChinaCropPhen1km. Compared with the intensive phenological observations from the Agricultural Meteorological Stations of China Meteorological Administration, the dataset had a high accuracy with errors of retrieved phenological date less than 10 days, and represented the spatiotemporal patterns of the observed phenological dynamics at site scale fairly well. The well-validated dataset can be applied for many purposes including improving agricultural system or earth system modelling over a large area.

**DOI of the referenced dataset:** https://doi.org/10.6084/m9.figshare.8313530 (Luo et al., 2019).

## 1 Introduction

Phenology is a key indicator of vegetation growth and development and plays an important role in vegetation monitoring (Qiu et al., 2015;Tao et al., 2017;Zhong et al., 2016). Accurate information on the timing of key crop phenological stages is critical for determining the optimal timing of agronomic management options, reliable simulations of crop growth and yield, and analyzing the plant response to climate change (Bolton and Friedl, 2013;Brown et al., 2012;Chen et al., 2018a;Sakamoto et al., 2013;Sakamoto et al., 2010;Wang et al., 2015;Zhang and Tao, 2013).





Field phenological observations are time- and money-consuming. And the observational stations are limited and distributed sparsely. Therefore, the field phenological observations can't meet the requirements of many purposes such as vegetation monitoring for remote areas with sparse observations and the grid-based earth system simulations. The satellite-based observations with a wide spatial coverage and short revisit times have become a powerful method to monitor vegetation growth and obtain vegetation information at regional and global scales. Previous studies have mainly used a vegetation index (VI) to

extract crop phenology. For examples, Pan et al. (2015) presented a method to construct Normalized Difference Vegetation Index (NDVI) time-series dataset derived from HJ-1 A/B CCD and extract phenology parameters. Zeng et al. (2016) detected corn and soybeans phenology with Moderate Resolution Imaging Spectroradiometer (MODIS) 250-m Wide Dynamic Range Vegetation Index (WDRVI) time-series data. Cao et al. (2015) developed an adaptive local iterative logistic fitting method to fit time-series of Enhanced Vegetation Index (EVI) derived from MODIS and estimate green-up date of spring vegetation.

Sakamoto (2018) refined the Shape Model Fitting method to estimate the timing of 36 crop-development stages of major U.S. crops from MODIS WDRVI time-series data. Crop phenology detected by these studies is relatively accurate. Nevertheless, it cannot be ignored that the VI are overly dependent on the band characteristics of sensors (Atzberger et al., 2014). By contrast, the Leaf Area Index (LAI) is more robust across diverse sensors and more sensitive than VI to large amounts of vegetation (Verger et al., 2016). In addition, previous studies focused on only very limited areas or very few crops due to the high diversity

and complexity of agricultural planting structures (Liao et al., 2019;Liu et al., 2017;Xu et al., 2017;Wang et al., 2012).

It is urgently required to acquire the gridded phenological datasets over a long-term period at a national scale as it is the basis for a large-scale agricultural system or earth system simulation. For example, crop model can simulate crop growth, development and predict crop yields. However, its applications to a large area are limited by the lack of accurate and spatially heterogeneous crop growth information (Curnel et al., 2011;Dorigo et al., 2007;Tao et al., 2009;Jin et al., 2018). According to

some previous studies, it could improve the accuracy of model estimation at large scale by assimilating reliable remote sensing data into crop growth models (Bolten et al., 2010;Nearing et al., 2012;Ines et al., 2013;Chen et al., 2018a;Huang et al., 2015;Zhou et al., 2019;de Wit and van Diepen, 2007). Among the state variables used in the assimilation, phenology is one of essential variables because of its critical roles in affecting dry matter accumulation and distribution during the growing stages and reflecting crop periodic biological changes influenced by various environmental conditions (e.g., climate) (Jin et al.,

2018;Zheng et al., 2016).

In this study, using a remotely sensed Global Land Surface Satellite (GLASS) LAI product (2000-2015) (Xiao et al., 2014), we aim to 1) choose the most suitable smoothing method to reduce the noise of the LAI time-series for different crops and regions; 2) detect the phenological information of three staple crops (i.e., maize, rice and wheat) at 1-km spatial resolution across China, and evaluate its accuracy by comparing with the observed data at Agricultural Meteorological Stations (AMS)

of China Meteorological Administration (CMA); 3) explore the spatial patterns of different phenological stages. The resultant remote sensing LAI-based crop phenology dataset with 1-km spatial resolution across China (ChinaCropPhen1km) will benefit to understand crop phenological dynamics, climate change impacts and adaptations, and agricultural system modelling over a large area, temporally and spatially (Luo et al., 2019).



## 2 Data and methods

### 2.1 Study area

The study areas are across the mainland of China, possessing of complex environments and crop planting structures, diverse cropping intensity and cultivation habits (Fig. 1) (Piao et al., 2010;Zhang et al., 2014a). Rice, wheat and maize are the three staple crops in China, together accounting for 59% of the total planting area and 92% of the grain yield in 2017. Roughly half of the cropland in China is multi-cropped, such as the single cropping system of spring maize in Northeast China, the double cropping system of wheat-maize in the North China Plain, and the rotation system between early-rice and late rice in Southern China (Frolking et al., 2002).

### 2.2 Data

#### 2.2.1 The GLASS LAI data

An improved MODIS-based LAI dataset (GLASS LAI) from 2000 to 2015 was from Liang et al. (2013) (http://glass-product.bnu.edu.cn/?pid=3&c=1). The GLASS LAI product was generated with general regression neural networks (GRNNs) trained by the fused LAI from MODIS and CYCLOPES LAI products and the reprocessed MODIS reflectance of the BELMANIP sites during the period 2001-2003 (Liang et al., 2013). Many studies have indicated that GLASS LAI (8-day composites of 1-km spatial resolution) was more temporally continuous and spatially complete than the other LAI products. It has been applied to vegetation monitoring and crop model assimilation (Xiao et al., 2014;Chen et al., 2018a).

#### 2.2.2 Phenology observation

The crop phenology observation records from 2000 to 2013 of maize, rice, and wheat crops were obtained from AMS, which were governed by CMA (https://data.cma.cn/). Such phenology information was observed and recorded by well-trained agricultural technicians in the experimental field, and then checked and managed by the Chinese Agricultural Meteorological Monitoring System (CAMMS). In this study, we selected the agrometeorological stations with more than 10 years of records of key phenological dates, including green-up date, emergence date, transplanting date, V3 stage (i.e., early vegetative stage of maize when the third leaf is fully expanded), heading date, and maturity date, for the three crops. Totally, there were 436 stations across main crop-cultivated areas in China (Fig. 1).

#### 2.2.3 Other data

The 1-km National Land Cover Dataset (NLCD) of China was provided by Data Centre for Resources and Environmental Sciences, Chinese Academy of Sciences (http://www.resdc.cn/Default.aspx), which also included several epochs of land use datasets including 2000, 2005, 2010 and 2015 (Liu et al., 2005;Liu et al., 2014).





## 2.3 Methods

The method to retrieve the phenological information of three staple crops at national scale is presented schematically in Fig. 2. The data processes are as follows: 1) data preprocessing, 2) selecting the cropland sample grid to determine the suitable smooth method, 3) determining the optimal filter-based phenology detection (OFP) approach, 4) retrieving the phenological information of three crops at 1-km pixel across China.

### 2.3.1 Data preprocessing

Due to the differences among these datasets on projected coordinate system, firstly, we projected or re-projected all raster data to "Asia North Albers Equal Area Conic" by using the Projection Raster tool in ArcGIS. Then, we combined 46 annual GLASS LAI images together and used a China provincial administrative vector map to mask images by province. Finally, a LAI time-series was created for each pixel for further applications.

### 2.3.2 Methods chosen to smooth LAI products

Previous studies have proposed different smoothing methods to reduce the noise of GLASS LAI time series, and found the OFP method varied by studied times, areas, and objectives (Zhao et al., 2016;Wang et al., 2018). Three popular methods were chosen in the study to smooth the LAI time-series curves, including the Double Logistic (DL) method, Savitzky-Golay (S-G) filter method, and Wavelet-based filter (WF) method.

#### 2.3.2.1 Double logistic (DL) method

Double logistic is a method of merging local fitting parts to obtain the overall fitting result (Jonsson and Eklundh, 2004). In the local fitting process, the Double logistic function can be expressed as:

$$g(t; x_1, \ldots, x_4) = \frac{1}{1 + exp(\frac{x_1 - t}{x_2})} - \frac{1}{1 + exp(\frac{x_3 - t}{x_4})} \tag{1}$$

where, $x_1$ determines the position of the left inflection point while $x_2$ gives the rate of change. Similarly, $x_3$ determines the position of the right inflection point while $x_4$ gives the rate of change at this point.

#### 2.3.2.2 Savitzky-Golay (S-G) filter method

Based on locally adaptive moving window, Savitzky-Golay (S-G) filtering method can be used to smooth data and suppress disturbances with a local polynomial regression model (Savitzky and Golay, 1964). Algorithm can be summarized as follows:

$$LAI_j^* = \frac{\sum_{i=-n}^{i=n} C_i LAI_{j+i}}{N} \tag{2}$$

where, $LAI_{j+i}$ represents the original LAI value, $LAI_j^*$ is the smoothed LAI value, j is running index of the LAI time series, $C_i$ is the coefficient of the i-th LAI value, n is the half-width of the smoothing window, and N is the width of the moving window





to perform filtering (2n+1). The width of the moving window—N, not only determines the degree of smoothing, but also

affects the ability to follow a rapid change. We selected three windows width (3, 4, 5) to identify a better width for different

crops and regions.

**2.3.2.3 Wavelet-based filter (WF) method**

Wavelet-based filter method can reduce noise with reflecting the periodicity of seasonal vegetation change (Sakamoto et al.,

2005). The input signals $f(x)$ is transformed in the wavelet transform as follows:

$$Wf(a,b)_i = \frac{1}{\sqrt{a}}\int \varphi(\frac{x-b}{a})f(x)dx \qquad (3)$$

where a is a scaling parameter, b is a shifting parameter, and $\varphi$ implies a mother wavelet.

The advantage of the WF method is that it can maintain the time components of time-series data and hardly distort signals.

The input signals $f(x)$ is decomposed to linear combinations of wavelet functions in the multi-resolution approximation:

$$f(x) = \sum_{i=1}^{j}[f(x)_i + g(x)_i] \qquad (4)$$

where $f(x)_i$ implies the approximate expression in level i, and $g(x)_i$ implies the high-frequency components in level i. We

used three types of mother wavelets: Daubechies (1988) (order=3–24), Coiflet (order=1–5), and Symlet (order=4–15) in the

study.

**2.3.3 Methods to detect the phenological information**

The methods to detect remote-sensing-based phenology can generally be classified into three groups: inflexion-based method

(Chen et al., 2016), threshold-based method (Manfron et al., 2017), and methods based on the mathematical or geometrical

model fitting approach (Sakamoto et al., 2010). In this study, we used both inflexion- and threshold-based methods together

to detect phenology. Firstly, we defined the inflection and maximum points of LAI time-series as the specific timing of key

phenological stages for different crops (Fig. 3).

**2.3.3.1 Green-up date, emergence date, transplanting date and V3 stage**

We defined the date of inflection point (the first derivative increases continuously after this point or the second derivative

equals 0) of the LAI time-series curves as the green-up date of winter wheat, emergence date of spring wheat, transplanting

date of rice and V3 stage (early vegetative stage of maize when the third leaf is fully expanded) of maize (Sakamoto,

2018;Sakamoto et al., 2005;Sakamoto et al., 2010). Before the inflection point, the LAI values are kept low for a long time,

and then they start to increase continuously after this point.





### 2.3.3.2 Heading date

Heading date in the study was defined as the day when LAI reaching the maximum, as similar as some previous studies (Sakamoto et al., 2005;Chen et al., 2018b). That is to say, the maximum LAI points in the time-series curve are regarded as the heading dates.

### 2.3.3.3 Maturity date

When crops reach maturity, the physiological activity will change largely, leading to an abrupt decrease in LAI (Sakamoto et al., 2005;Chen et al., 2018b). Therefore, we regarded an inflection point in the LAI time-series curve, where the first derivative is negative with the largest absolute value, as the maturity date.

### 2.3.4 Determining the optimal filter-based phenology detection approach (OFP)

Based on the observations around the nearest AMS, we needed firstly to determine the restricted time windows responding to each key phenological stage for different crops. Then we sampled randomly 1000 grids every year in each province from the grids where the land use data was identified as cropland and retrieved the key phenological stages in the sampling grids according to the three smoothing methods and the above definitions of key stages. To determine the OFP approach in each province, we identified the inflection points and maximum value point of each LAI time-series curve at each grid within the restricted time windows. After detecting phenological information of the cropland sample grids, we calculated the RMSE values between the estimated phenological dates and observed dates, and averaged these RMSE values for each crop at a provincial scale. Finally, we chose the most suitable smoothing method for different crops in each province with minimum RMSE.

### 2.3.5 Retrieving the phenological information at 1-km pixel across China

After removing the grids from where the land use data was identified as non-cropland, we then obtained cropland grids where the phenological information will be detected. Then, the most suitable smoothing method for different crops in each province were applied to reconstruct the LAI time series at 1-km grid scale. Finally, we detected the key phenological dates based on OFP approach and regarded the grids that the three key phenological stages (mentioned in 2.3.3) could be simultaneously identified as the crop- cultivated grid for each crop. Additionally, to evaluate the accuracy of the estimated phenological dates at a national scale, we calculated the mean of phenological dates detected from each crop pixels around corresponding AMS and compared them with the corresponding observations by RMSE.





## 3 Results and Discussion

### 3.1 Comparisons of different smoothing methods

The smoothed time profiles of LAI generated by different smoothing methods are shown in Fig. 4. Both S-G filter and WF method can smooth LAI time-series well. That is to say, the generated time profiles of LAI match well with the seasonal tendency of the observed LAI time-series in the field. In addition, both methods can clearly characterize the local changes in the time component and maintain the time components of LAI time-series data. Although DL method performs poorly for smoothing LAI time series of the double-season crops, it's still reliable for single-season crops. These findings were consistent with those in some previous studies (Zhu et al., 2012;Sakamoto et al., 2005;Qiu et al., 2016).

We further compared mean RMSE of different smoothing methods and selected the most suitable smoothing method with minimal mean RMSE for different provinces and crops (Table 1). If the RMSE values were same, we also compared the number of crop grids according to different smoothing methods, and selected the suitable method which had identified a larger number of crop girds. It is noted that the number of identified grids differ considerably even with same RMSE values. Totally, S-G filter was an overwhelming smoothing way for 95% crops and provinces, followed by WF and DL method.

We ascribed the great performance of S-G to two reasons as follows. 1) One is its scientific smoothing principle: S-G filter applies an iterative weighted moving average filter to the time series, which can replace the noise data as well as keep the fidelity (Geng et al., 2014). By contrast, WF decomposes the time series into scaled and shifted wavelets to acquire time-localization of a given signal (Qiu et al., 2014). DL uses a series of parameters to fit the time series (Beck et al., 2006). 2) The other is that S-G is more suitable for GLASS LAI. S-G can catch the local variations—e.g. the bimodal curve characteristics from double cropping rice and the rotation of winter wheat and summer maize (the median columns of Fig.3) — in time series and perform best for data without extreme noise such as GLASS LAI (Eklundh and Jönsson, 2015). DL is more useful for data with much noise, however, fails to catch local changes due to being unfit for data with double peaks. WF is also a powerful tool for processing non-stationary and noisy signals such as VI time-series rather than GLASS LAI (Rouyer et al., 2008; Sakamoto et al., 2006). Therefore, S-G is the most suitable for the complex cultivating systems across whole mainland of China. We also attributed the excellent performance of S-G to the phenological extraction rules established in this paper, and the goal of accurately extracting the crop cultivation grids, as well as key phenology stages. For example, WF smoothing method might eliminate pseudo inflexion points that may not be pseudo due to the uncertainty of GLASS LAI data sometimes, and misidentify non-crop grids by inflexion- and threshold-based methods consequently resulting in very few crop grids identified (Qiu et al., 2016).

### 3.2 Validation of the phenological data

The comparison between retrieved phenological dates and phenological observations of each crop from 2000-2015 at national scale showed that all retrieved and observed dates were closely and averagely distributed 1:1 line for three crops (Fig. 5). Additionally, the RMSE values of retrieved phenological dates were consistently less than 10 days (Table 2). The RMSE



averages of three key dates for rice were around 5.3 days, followed by wheat (5.5 days) and maize (6.7 days), corresponding to the related $R^2$ of 0.98, 0.97 and0.97, respectively.

As for the differences among crops, the retrieved accuracy of maize phenological stages was consistently the worst, with the biggest RMSE and errors ($\geq\pm10$ days), and the lowest errors ($\leq\pm10$ days) and $R^2$. We ascribed the lower accuracy of maize phenology to the wider spatial heterogeneity environment and the complex rotation planting system relative to the other two crops (Qiu et al., 2018). The highest accuracy of rice phenology also supported the accuracy impact of complex planting system because paddy field is unfit for dryland crops such as maize, wheat, soybean, and other coarse cereals (Dong et al., 2015).

More interestingly, the retrieved accuracy of three crops decreased as crop growing and developing up to maturity periods (Table 2), with the average RMSEs ranging from 3.7 to 7.2 days. The highest accuracy (RMSE=2.8, error=0.5%) was found for the green up/emergence stages of wheat, while accuracy degrees of maturity stage for each crop were the lowest (average RMSE=7.2, error=19%). The reasonable explanation might be relative weaker interfere from other vegetation because the green-up/emergence stage occurs most early during plant growing period (some 80 DOY Table 3). With the land surface

greening up, more and more information on plant growing statuses will be shot by satellites, which consequently mix with the crops' information and interfere to retrieve accurately the phenological stages of crops. Of course, the interfering from anthropological activities should not ignored with climate warming.

Nevertheless, overall the retrieved phenological dates for the three crops are in strong correspondence with the observational dates ($R^2 > 0.95$) and their relationships are statistically significant ($p < 0.01$). Meanwhile, the growing status of other plants

(or rotation crops e.g. wheat-maize, maize-soybean) and the influence of other noises will lead to deviations of the remote-sensing LAI curve and the actual observed curve in the field. The noises also include other factors, e.g. weather conditions, farmers' behaviors, etc... However, the uncertainty does not exclude the applicability of our method to retrieve key phenological stages of crops, especially retrieving relative higher resolution phenological information based on mature remote-sensing products at a large spatial scale.

**3.3 Spatiotemporal patterns of key phenological stages from 2000 to 2015**

We showed the annual averages of each key crop growth stage to indicate their spatiotemporal patterns due to the similarity in inter-annual patterns for a certain crop over the 16 years (Figs.6-7, Table 3, and Fig. S1-S4). Besides summarizing the key stages by crops and sub-regions, we also calculated three crop growth periods, including VGP (vegetative growth period), RGP (reproductive growth period) and $GP_W$ (whole growth period) to interpret their patterns (Fig.8, Table 3). Among the five

sub-regions with rice cultivation, the sub-region III was the most complex because three types of rice were cultivated there (Fig.6-a; Fig.7-a). The single-rice in the sub-region III was generally cultivated in the northern parts of four provinces (i.e., Anhui, Jiangsu, Zhejiang and Hubei), which was characterized by three key stages occurring latest (DOY 159~265) than other three single-rice sub-regions (I, II, IV). Moving from the south (IV) to north (I) (excluding the sub-region III because of cultivation of double-rice), single-rice wasn't transplanted in sequence as expected. In the sub-region II, it was transplanted

latest (DOY 154) but had relatively early maturity dates (DOY 255), resulting in the shortest growing period (101 days) (Fig.8-





a). On the contrary, in the sub-region IV, single-rice was transplanted earliest but had the maturity occurring latest, resulting in the longest growing period of 130 days (Figs.6-a, 7-a, 8-a). In the sub-region V where only double rice was cultivated, early-rice was transplanted earlier (DOY 99), maturity dates of late rice occurred later (DOY 310), and consequently resulting in longer growing periods (97 and 101 days for corresponding early and late-rice) than those in the sub-region III with double-rice cultivation (80 and 84 days) (Table 3).

As for wheat, green-up & emergence dates ranged most widely (DOY 30~128) than other crops. Winter wheat in the sub-regions II and IV had earlier green-up dates, while spring wheat in the sub-regions of I and III had later emergence dates (Table 3, Figs.6-b, 7-b). Moreover, along with the latitudes from the north to south (excluding the sub-region III because of the sparsest wheat cultivation there), the first key dates became earlier but with shorter growth periods (106, 93, 92 days for I, II and IV) due to the sufficient temperature and light in the sub-region IV (Yu et al., 2012) (Fig.8-b). Interestingly, the heading and maturity dates in the three sub-regions showed consistently the same spatial patterns as that of the first stage with latitudes decreasing (Figs.6-b, 7-b).

Both spring and summer maize types were concurrently cultivated in the sub-regions III and IV, while only one of them was cultivated in the sub-regions I and II, the main planting areas of northern China (Figs. 6-c, Fig.7-c, Table 3). V3 of summer maize was approximately 43 days later than that of spring maize (DOY 161 vs. 117), but their maturity dates were very close (DOY 259 vs. 245), which thus caused a shorter growth period for summer maize, especially for the sub-regions II and III (some 84 days) (Fig.8-c). Additionally, in three sub-regions (I, III, IV) for spring maize, like wheat, the spatial patterns of the three key stages for maize were similar in spatial patterns with latitude increasing. Finally, the key dates and periods were the most variable in the sub-region IV (Figs.7-c, 8-c).

In sum, the spatial patterns of key phenological stages varied by crops and cultivated ways. In addition, early rice and single cropping rice in the sub-region III, wheat in the sub-region III, and maize and rice in the sub-region IV showed a larger variability than others due to the mixed planting of heterogeneous varieties of the same crop. Many factors could have impacts on crop phenology, such as climate, environment, farmer's behaviors, technological development, and human activities (Liu et al., 2016;Liu et al., 2018). Different from natural ecosystems such as wild forest or grassland, three main crops cultivated across the mainland of China didn't reach greening-up or flowering dates in sequence with latitude, especially for rice (Zhang et al., 2015;Tao et al., 2014;Zhang et al., 2014b). Moreover, climate conditions did have impacts on crop phenology. For example, increased temperature had advanced heading and maturity date of crops in China (He et al., 2015;Tao et al., 2014). At the same time, crop management activities, such as cultivar shift and the adjustment of planting and harvesting date, had affected crop phenology largely (Tao et al., 2006;Tao et al., 2013).

### 3.4 The changes in three key phenological dates and growth periods from 2000 to 2015

To interpret the changes of the three key phenological dates and growth periods from 2000 to 2015, we analyzed their trends at pixel scale and summarized the grids with a significant trend ($p<0.1$) according to crops and sub-regions (Figs 9-10, Table 4). We found more positive trends, with 0.78 days/year for 70% summarized medians, but fewer negative ones, with -0.69



days/year and 30% medians. This suggests that phenological dates have been delayed. Specifically, the proportion of pixels that had a positive trend is 92% for wheat (Fig.9-b), 75% for rice (Fig.9-a), and 50% for maize (Fig.9-c).

For rice, transplanting dates were consistently advanced by -0.64 days/year for early rice and single-rice, and delayed by 0.84 days/year for late rice in most areas. Maturity dates became later by 1.23 days/year, but heading dates had less changes. In addition, double rice in the sub-region III showed less variable than that in the sub-region V (Fig.9-a). By contrast, the first stages (i.e., green-up & emergence/V3) delayed by 0.88 days/year consistently for almost all the wheat cultivation areas (Fig.9-b). Maize in the sub-region II, and wheat in the sub-region I and II (Fig.9-b), had an opposite trend to that of rice (Fig.9, Table 4). Moreover, the changes in the three stages showed less variable in the sub-region II, the main planting areas for both dryland crops. Among all the crops and growth stages, maize in the regions III and IV had consistently negative trends with exception of maturity dates in the sub-region IV.

Compared with the significant changes in phenological dates, the duration of phenological periods changed in less pixels (< 30%) (Table 4). More pixels with positive trends, with 1.25 days/year for 66.7% medians, were identified than those with negative trends, with -0.97 days/year for 32.3% medians, implying a commonly prolonged growth periods during the study period. 95.8% of the medians were positive for rice, while 75% of the medians were negative wheat. The changes of maize growth periods were similar to those of its phenological dates.

The duration of growth periods was prolonged, especially for the whole growth period (GPw), which was consistently observed for rice cropping systems, except for early rice in the sub-region V. In addition, the duration of VGP for single rice in the sub-region I had weaker trends (Fig. 10-a). On the contrary, almost all the wheat growth periods were shortened except for winter wheat in the sub-region V, especially for spring wheat in the sub-region I (Fig.10-b). Additionally, in term of growth period duration, maize had the similar changes as wheat in the sub-regions I and II. Changes in growth period duration were different for spring (shortened) and winter (prolonged) wheat, and for both maize types between the sub-region III and IV (Fig.10-c). The results are well supported by some previous studies based on the intensive observations at site scale (Tao et al., 2013;Tao et al., 2014;Tao et al., 2012;Zhang et al., 2014b).

## 4 Data availability

The derived crop phenological dataset for three staple crops in China during 2000-2015 is available at https://doi.org/10.6084/m9.figshare.8313530 (Luo et al., 2019).

## 5 Conclusion

In the present study, we proposed a method to retrieve 1km-grid crop phenological dataset for three main crops from 2000 to 2015 based on GLASS LAI products. First, we compared three common smoothing methods and chose the most suitable methods for different crops and regions. The results showed that S-G was the most frequently chosen method as it not only



could well smooth the time series but also keep the fidelity. Next, we developed an OFP approach which combined both inflexion- and threshold-based method to detect the key phenological stages of three staple crops at spatial resolution of 1km across China. Finally, we established a high resolution gridded-phenology product for three staple crops in China during 2000-2015, i.e., ChinaCropPhen1km.

The ChinaCropPhen1km dataset has been well validated using the intensive phenological observations of AMS. It can reflect the spatial differences in the local climatic and management factors. Thus, this first high-resolution crop phenological dataset
can be applied for many purposes, including understanding land surface phenological dynamics, investigating climate change impacts and adaptations, and improving agricultural system or earth system modelling over a large area, temporally and spatially.

**Author contribution**

Zhao Zhang and Yi Chen designed the research. Yi Chen and Ziyue Li collected datasets. Yuchuan Luo implemented the
research and wrote the paper; Zhao Zhang and Fulu Tao revised the manuscript.

**Competing interests**

The authors declare that they have no conflict of interest.

**Acknowledgements**

This study was funded by National Basic Research Program of China (41571493, 41571088, 31561143003), and State Key
Laboratory of Earth Surface Processes and Resources Ecology.

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



**Table 1: Mean RMSE (in parenthesis) of the most suitable smoothing method for different regions and crops.**

|  | Winter wheat | Spring wheat | Summer maize | Spring maize | Single-rice | Double rice |
|---|---|---|---|---|---|---|
| Anhui | SG-3 (3.53) |  | SG-4 (6.81) |  | SG-5 (5.67) | SG-3 (6.69) |
| Beijing | SG-3 (5.46) |  | SG-3 (8.06) |  |  |  |
| Chongqing | SG-3 (7.09) |  |  | SG-3 (8.06) | SG-3 (3.46) |  |
| Fujian | SG-3 (8.90) |  |  |  | SG-3 (6.36) | SG-3 (6.02) |
| Gansu | SG-3 (5.43) | SG-3 (7.69) | SG-5 (7.63) | SG-3 (8.34) |  |  |
| Guangdong |  |  |  |  |  | SG-3 (6.38) |
| Guangxi |  |  |  | SG-3 (8.17) |  | SG-3 (7.25) |
| Guizhou | SG-3 (6.59) |  | SG-3 (9.77) | SG-3 (9.11) | SG-3 (7.62) |  |
| Hainan |  |  |  |  |  | db8 (2.33) |
| Hebei | SG-3 (4.53) | SG-3 (6.35) | SG-3 (4.58) | SG-3 (5.42) | SG-3 (6.32) |  |
| Heilongjiang |  | SG-5 (6.38) |  | SG-5 (6.72) | SG-5 (4.73) |  |
| Henan | SG-3 (4.22) |  | SG-4 (5.01) |  | SG-5 (3.48) |  |
| Hubei | SG-3 (5.65) |  |  | SG-3 (7.89) | SG-3 (5.87) | SG-3 (5.89) |
| Hunan |  |  |  | SG-3 (8.10) | SG-3 (5.03) | SG-3 (7.64) |
| Jiangsu | SG-3 (5.12) |  | SG-4 (6.44) | db4 (8.67) | SG-5 (6.85) |  |
| Jiangxi |  |  |  | SG-3 (7.95) | SG-3 (7.80) | SG-3 (8.12) |
| Jilin |  | SG-4 (7.73) |  | SG-5 (5.84) | SG-5 (6.30) |  |
| Liaoning |  |  |  | SG-3 (4.91) | SG-3 (6.39) |  |
| Inner Mongolia |  | SG-3 (7.85) |  | SG-5 (5.55) |  |  |
| Ningxia | SG-3 (6.18) | SG-3 (6.74) | SG-3 (7.50) | SG-5 (6.33) | SG-5 (8.22) |  |
| Qinghai | db3 (6.73) | DL (7.56) |  | SG-5 (7.59) |  |  |
| Shandong | SG-3 (4.46) |  | SG-4 (4.55) |  | SG-5 (6.36) |  |
| Shanghai | SG-3 (5.01) |  |  |  | SG-3 (7.15) |  |
| Shannxi | SG-3 (4.04) | SG-3 (8.08) | SG-3 (4.09) | SG-3 (5.05) | SG-5 (7.57) |  |
| Shanxi | SG-3 (4.61) | DL (7.90) | SG-3 (5.45) | SG-5 (5.57) | SG-5 (8.84) |  |
| Sichuan | SG-3 (5.43) |  | SG-3 (7.43) | SG-3 (7.84) | SG-3 (5.51) |  |
| Tianjin | SG-3 (7.36) |  | SG-3 (8.17) |  |  |  |
| Xinjiang | SG-3 (6.93) | SG-3 (7.99) | SG-3 (7.11) | SG-3 (6.14) |  |  |
| Xizang | SG-3 (7.02) | SG-3 (7.12) |  |  |  |  |
| Yunnan | SG-3 (7.53) |  | SG-3 (8.45) | SG-3 (8.19) | SG-5 (7.53) | SG-3 (4.51) |
| Zhejiang | SG-3 (6.22) |  |  |  | SG-3 (6.35) | SG-4 (7.33) |





**Table 2: Mean RMSE between retrieved phenological dates and phenological observations.**

| Crop | Stage | RMSE (days) | Error ≤ ±10 days (%) | Error ⩾ ±10 days (%) | $R^2$ |
|------|-------|-------------|----------------------|----------------------|-------|
| Rice | Transplanting | 4.05 | 98.6% | 1.4% | |
| | Heading | 5.59 | 93.0% | 7.0% | 0.98 |
| | Maturity | 6.21 | 88.9% | 11.1% | |
| Wheat | Green up & Emergence | 2.82 | 99.5% | 0.5% | |
| | Heading | 6.54 | 86.4% | 13.6% | 0.97 |
| | Maturity | 7.18 | 81.7% | 18.3% | |
| Maize | V3 | 4.08 | 96.8% | 3.2% | |
| | Heading | 7.79 | 79.8% | 20.2% | 0.97 |
| | Maturity | 8.22 | 71.8% | 28.2% | |






**Table 3: Annual mean phenological dates and growth periods of different crops in each sub-region**

| Crop | Stage (period) | Sub-region | | | | |
|---|---|---|---|---|---|---|
| | | I | II | III | IV | V |
| Early rice | Transplanting (VGP) | | | 115.4 (60) | | 99 (69.3) |
| | Heading (RGP) | | | 175.2 (20.4) | | 168.2 (27.6) |
| | Maturity ($GP_w$) | | | 195.1 (79.7) | | 195.5 (96.6) |
| Late rice | Transplanting (VGP) | | | 204.2 (48.1) | | 209.4 (54.1) |
| | Heading (RGP) | | | 252.2 (35.5) | | 265.7 (46.5) |
| | Maturity ($GP_w$) | | | 287.7 (83.5) | | 310 (100.6) |
| Single rice | Transplanting (VGP) | 141.7 (75.5) | 154.1 (64.2) | 158.5 (62.5) | 130.4 (76.6) | |
| | Heading (RGP) | 217.2 (37.9) | 218.3 (37.1) | 221 (43.7) | 207 (53.4) | |
| | Maturity ($GP_w$) | 255 (113.5) | 255.3 (101) | 264.6 (106) | 260.4 (130) | |
| Wheat | Green up & Emergence (VGP) | 128 (65.1) | 51.8 (62.9) | 108.7 (57.5) | 29.6 (34.7) | |
| | Heading (RGP) | 187.6 (41.1) | 113.9 (29.6) | 165.7 (63.7) | 72.3 (57.1) | |
| | Maturity ($GP_w$) | 224.8 (106) | 143.3 (92.5) | 228.8 (121) | 128.7 (91.7) | |
| Spring maize | V3 (VGP) | 142.4 (71.9) | | 130.2 (79.6) | 104.3 (74.5) | |
| | Heading (RGP) | 214.3 (40.6) | | 209.9 (42.2) | 178.8 (59.7) | |
| | Maturity ($GP_w$) | 254.9 (113) | | 252.1 (122) | 238.5 (134) | |
| Summer maize | V3 (VGP) | | 173.1 (46.9) | 179.1 (47.1) | 129.3 (74.1) | |
| | Heading (RGP) | | 220.1 (37.5) | 226.2 (36.1) | 203.3 (53.3) | |
| | Maturity ($GP_w$) | | 257.5 (84.5) | 262.2 (83.1) | 256.6 (127) | |

Note: VGP means vegetative growth period, the difference between heading and transplanting/Green up & Emergence/V3 dates; RGP means reproductive growth period, the difference between maturity and heading dates; $GP_w$ means whole growth period, the difference between maturity and transplanting/Green up & Emergence/V3 dates. The numbers in the parentheses mean the annual mean growth periods.



**Table 4: The trend (days year$^{-1}$) of three key phenological dates and growth periods from 2000 to 2015**

| Crop | Stage (period) | Sub-region | | | | | | | | | |
|---|---|---|---|---|---|---|---|---|---|---|---|
| | | I | | II | | III | | IV | | V | |
| | | Trend | Psg | Trend | Psg | Trend | Psg | Trend | Psg | Trend | Psg |
| Early rice | Transplanting | | | | | -0.7 | 27.8 | | | -0.8 | 27.3 |
| | (VGP) | | | | | (0.6) | (19.8) | | | (1.5) | (11.8) |
| | Heading | | | | | -0.8 | 26.1 | | | 1.3 | 29.1 |
| | (RGP) | | | | | (1.3) | (15.5) | | | (1.0) | (14.5) |
| | Maturity | | | | | 0.7 | 23.3 | | | 1.6 | 29.6 |
| | (GP$_w$) | | | | | (1.3) | (19.4) | | | (1.7) | (14.3) |
| Late rice | Transplanting | | | | | 0.7 | 25.4 | | | 1.7 | 28.2 |
| | (VGP) | | | | | (0.01) | (15.6) | | | (0.01) | (44.3) |
| | Heading | | | | | 0.7 | 31.1 | | | 1.7 | 29.1 |
| | (RGP) | | | | | (1.0) | (16.5) | | | (2.2) | (13.7) |
| | Maturity | | | | | 1.1 | 32.6 | | | 2.4 | 35.4 |
| | (GP$_w$) | | | | | (1.1) | (15.8) | | | (2.3) | (13.4) |
| Single rice | Transplanting | 0.01 | 31.6 | -1.1 | 43.3 | 0.01 | 37.9 | -1.3 | 22.3 | | |
| | (VGP) | (-0.01) | (29.3) | (1.3) | (31.6) | (0.8) | (22.9) | (1.3) | (13.5) | | |
| | Heading | 0.01 | 38.7 | 0.01 | 32.4 | 0.7 | 31.6 | -0.01 | 19.1 | | |
| | (RGP) | (0.7) | (20.1) | (0.7) | (18.1) | (0.8) | (16.8) | (1.8) | (17.6) | | |
| | Maturity | 0.7 | 20 | 0.7 | 28.2 | 1 | 28.5 | 1.6 | 24.1 | | |
| | (GP$_w$) | (0.6) | (14.8) | (1.7) | (29.8) | (1.2) | (22.1) | (2.1) | (19.7) | | |
| Wheat | Green up & Emergence | 0.01 | 61.6 | 0.6 | 16.9 | 1.2 | 15 | 1.7 | 26.5 | | |
| | (VGP) | (-1.5) | (22.2) | (-0.4) | (12.8) | (-0.9) | (14.1) | (-2.0) | (12.1) | | |
| | Heading | -0.01 | 34.6 | 0.6 | 21.9 | 0.9 | 23.4 | 0.8 | 35 | | |
| | (RGP) | (0.01) | (13.1) | (-0.6) | (14.9) | (-0.8) | (12.7) | (2.3) | (14.8) | | |
| | Maturity | 0.01 | 30.3 | 0.01 | 22.1 | 1.4 | 17.3 | 1.9 | 42.8 | | |
| | (GP$_w$) | (-1.9) | (17.4) | (-0.7) | (12.1) | (-1.2) | (12.4) | (1.7) | (11.2) | | |
| Spring maize | V3 | 0.01 | 26.2 | | | -1.1 | 20.4 | -1.0 | 33.6 | | |
| | (VGP) | (-0.9) | (20.9) | | | (1.2) | (12.6) | (-1.3) | (15.1) | | |
| | Heading | -0.01 | 54.8 | | | 0.01 | 55.4 | -1.4 | 28.9 | | |
| | (RGP) | (0.9) | (20.1) | | | (0.9) | (12.8) | (2.0) | (16.5) | | |



| | | | | | | | | | |
|---|---|---|---|---|---|---|---|---|---|
| | Maturity | 0.01 | 26.4 | | | -0.5 | 31.4 | 0.7 | 31.3 |
| | (GP$_w$) | (-0.7) | (13.9) | | | (1.4) | (11.8) | (1.5) | (14.6) |
| Summer maize | V3 | | | 0.01 | 52.6 | -0.9 | 27.6 | -0.01 | 41.2 |
| | (VGP) | | | (-0.7) | (22.1) | (-1.0) | (12.1) | (1.2) | (11.8) |
| | Heading | | | 0.4 | 27.3 | -0.01 | 30.1 | 0.01 | 40.1 |
| | (RGP) | | | (-0.8) | (13.4) | (0.01) | (13.2) | (2.3) | (19.1) |
| | Maturity | | | 0.4 | 20.1 | -1.4 | 34.7 | 2.4 | 48.8 |
| | (GP$_w$) | | | (-1.0) | (16.7) | (-1.1) | (12.9) | (2.5) | (18.3) |

Note: The same meanings for VGP, RGP and GP$_w$ as Table 3; Psg (%) means the percent of grids showing significant trend at $p<0.1$ level; The numbers in the parentheses mean the statistic values of grids during three growth periods.


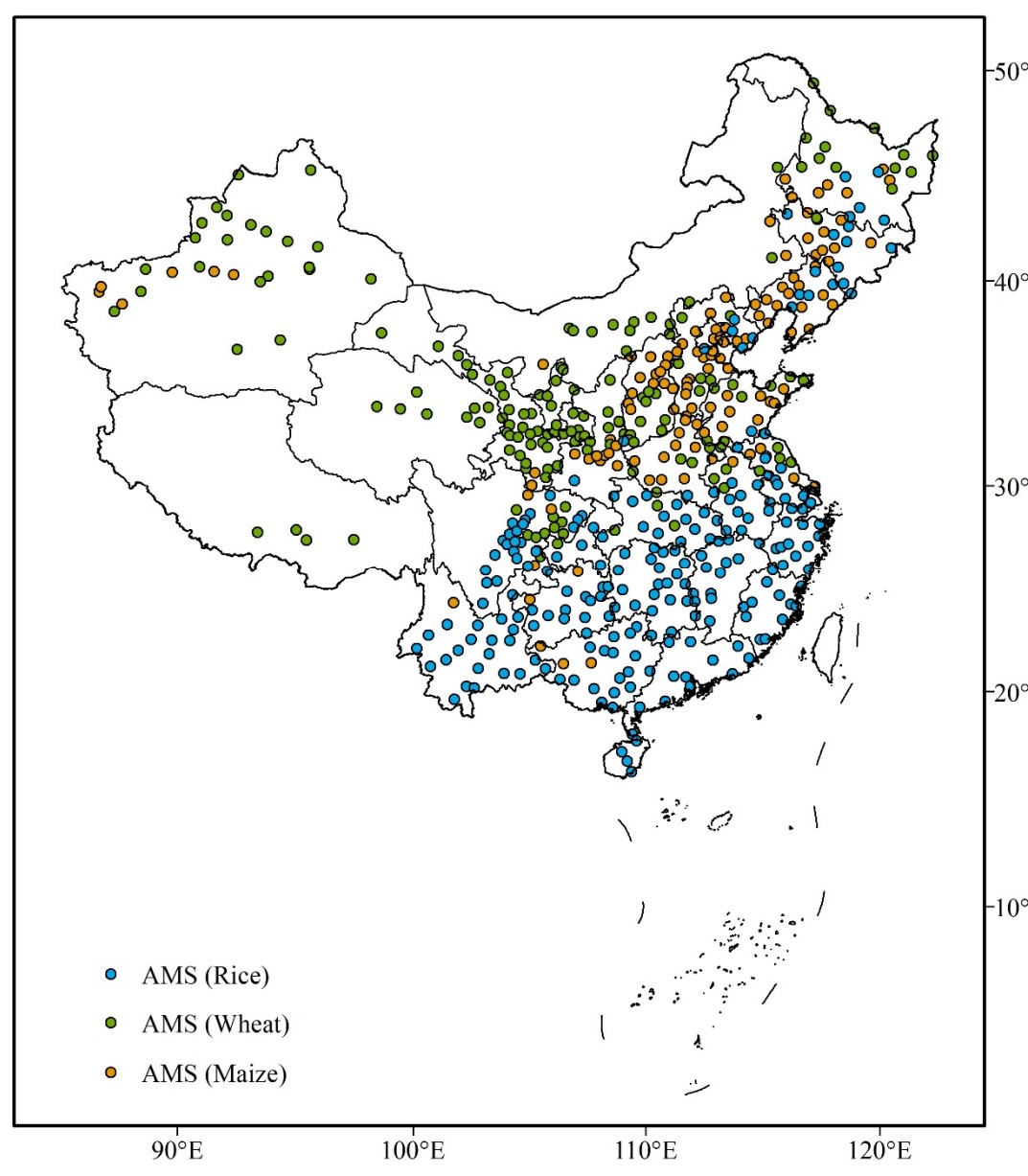

**Fig.1: The studied areas and the locations of Agricultural Meteorological Stations (AMS) of China Meteorological Administration.**





**Fig.2: Flow chart of procedures for data analysis and crop phenological dates identification.**



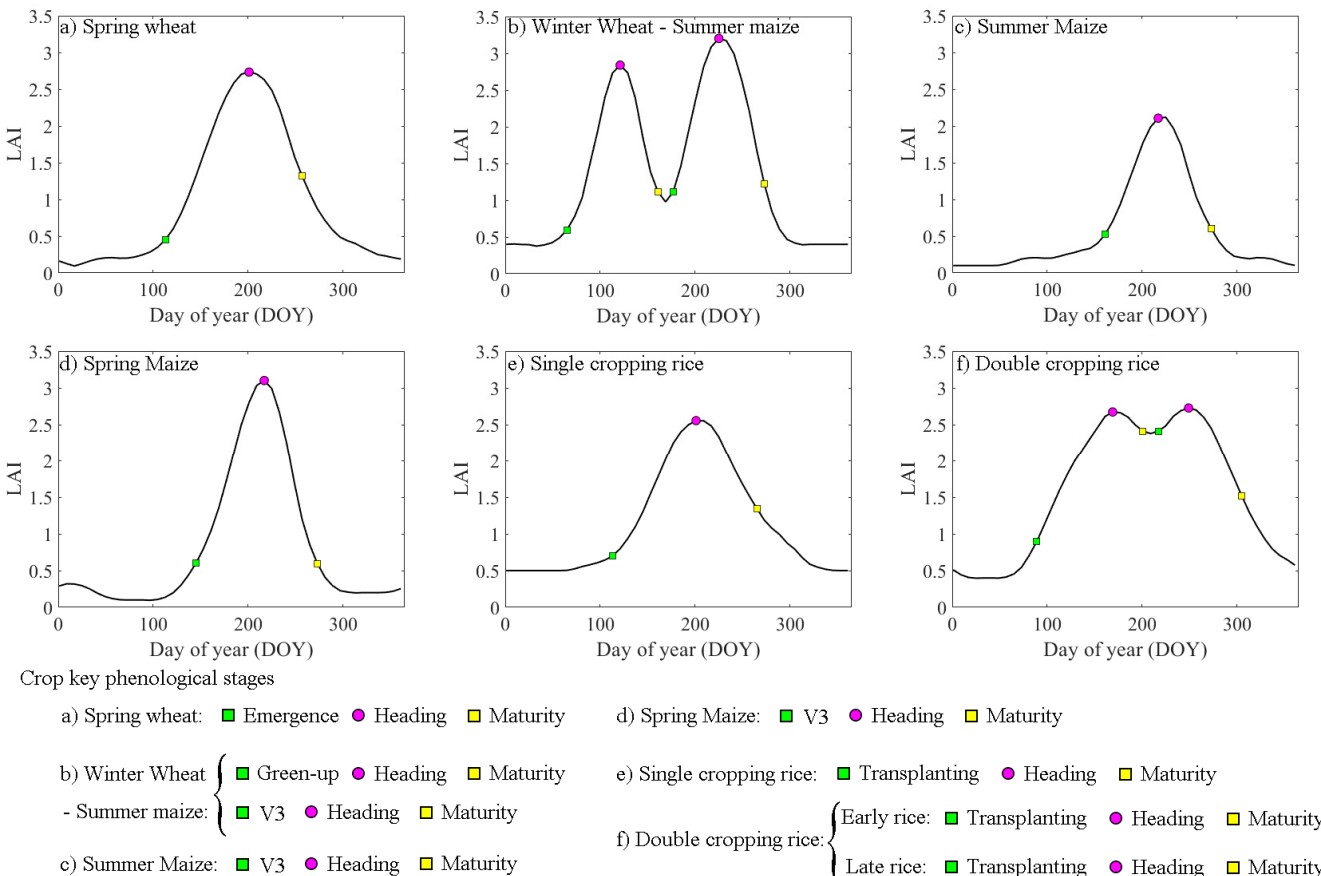

**Fig.3: Typical phenological curves for different crop cropping systems in China.**




**Fig.4: Comparisons of different smoothing methods for different cropping systems.**

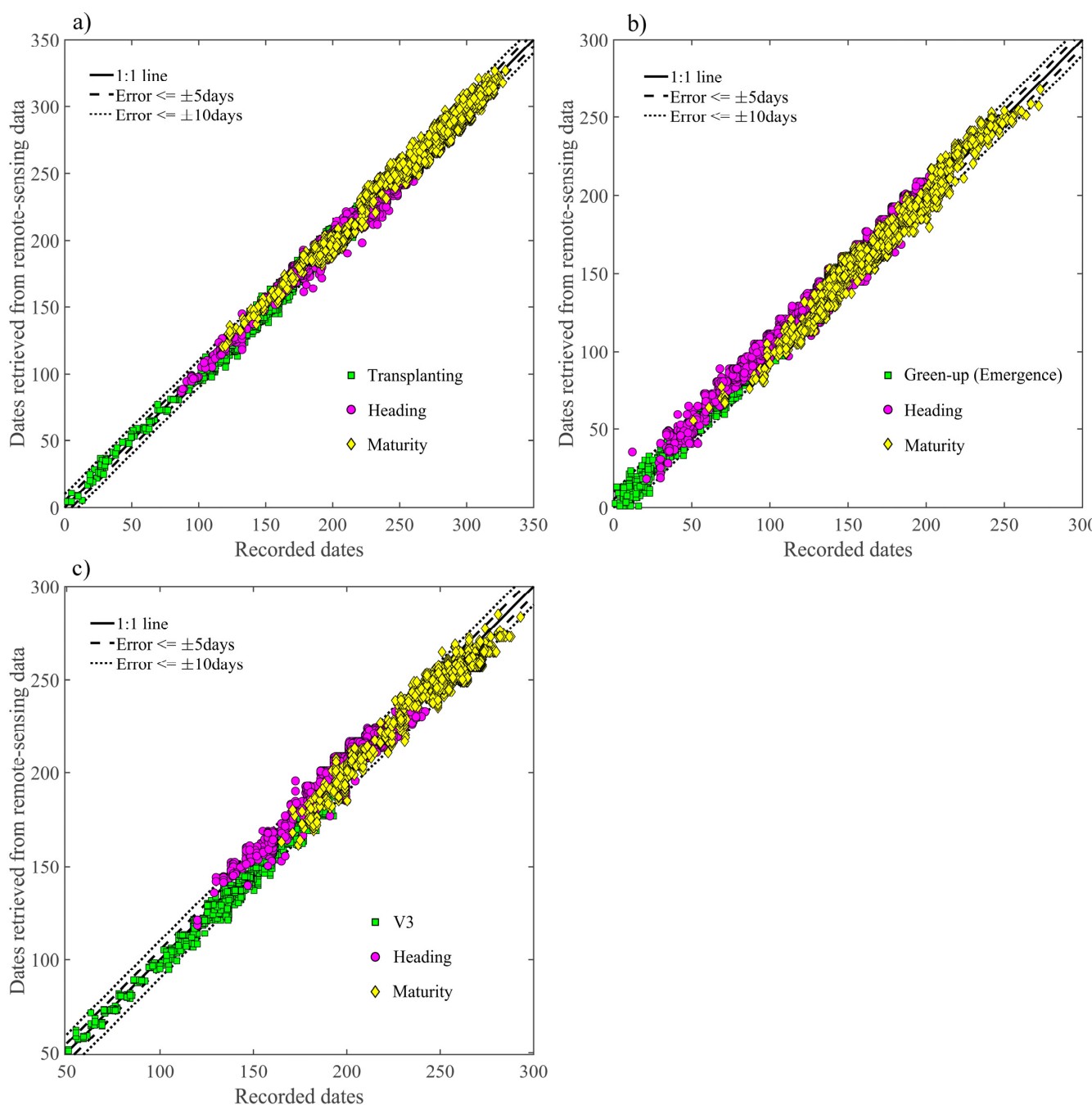

**Fig.5: Comparisons between retrieved and observed phenological dates for rice (a), wheat (b), and maize (c).**





**Fig.6: Spatial patterns of annual averages of three key phenological dates during 2000~2015 for rice (a), wheat (b), and maize (c).**

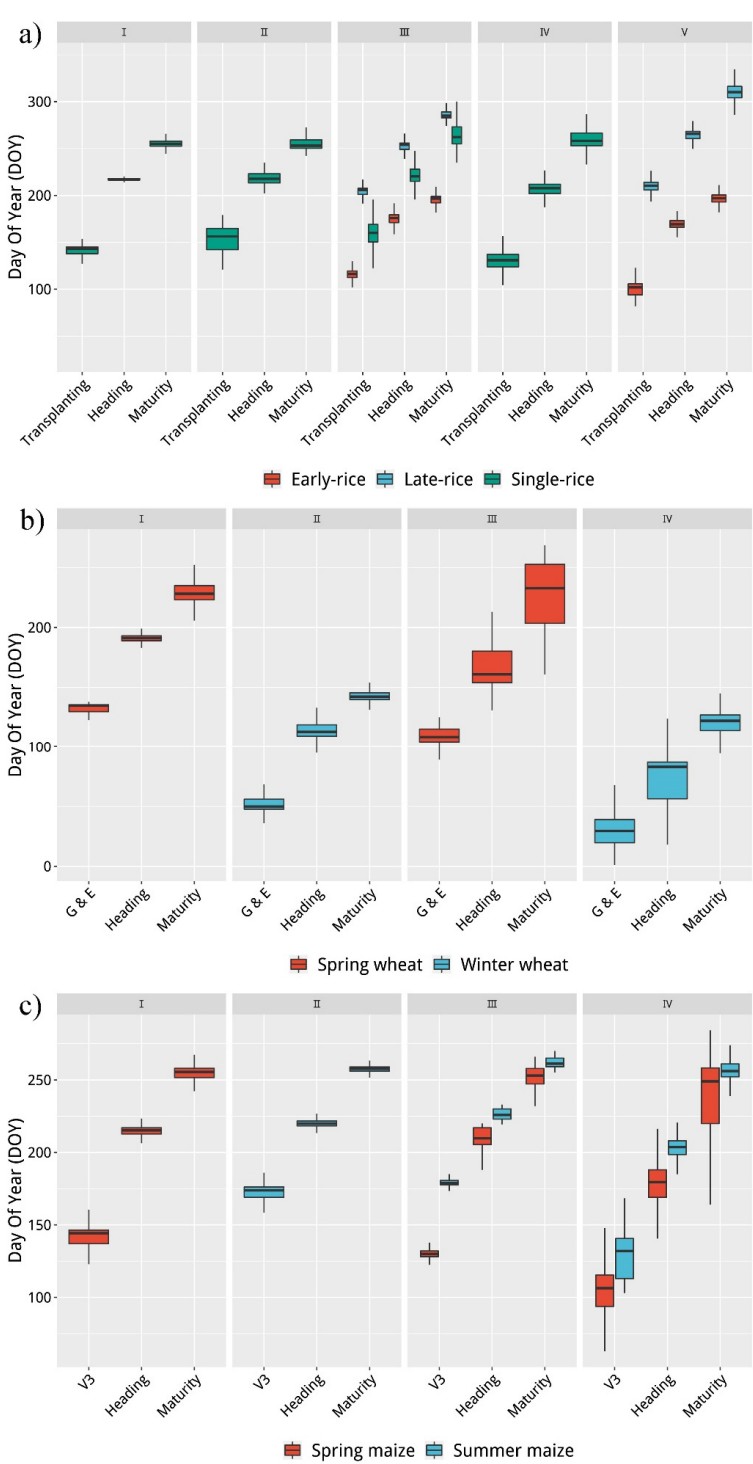

**Fig.7: Box plots of three key phenological dates by crop and sub-regions during 2000~2015 for rice (a), wheat (b), and maize (c).**



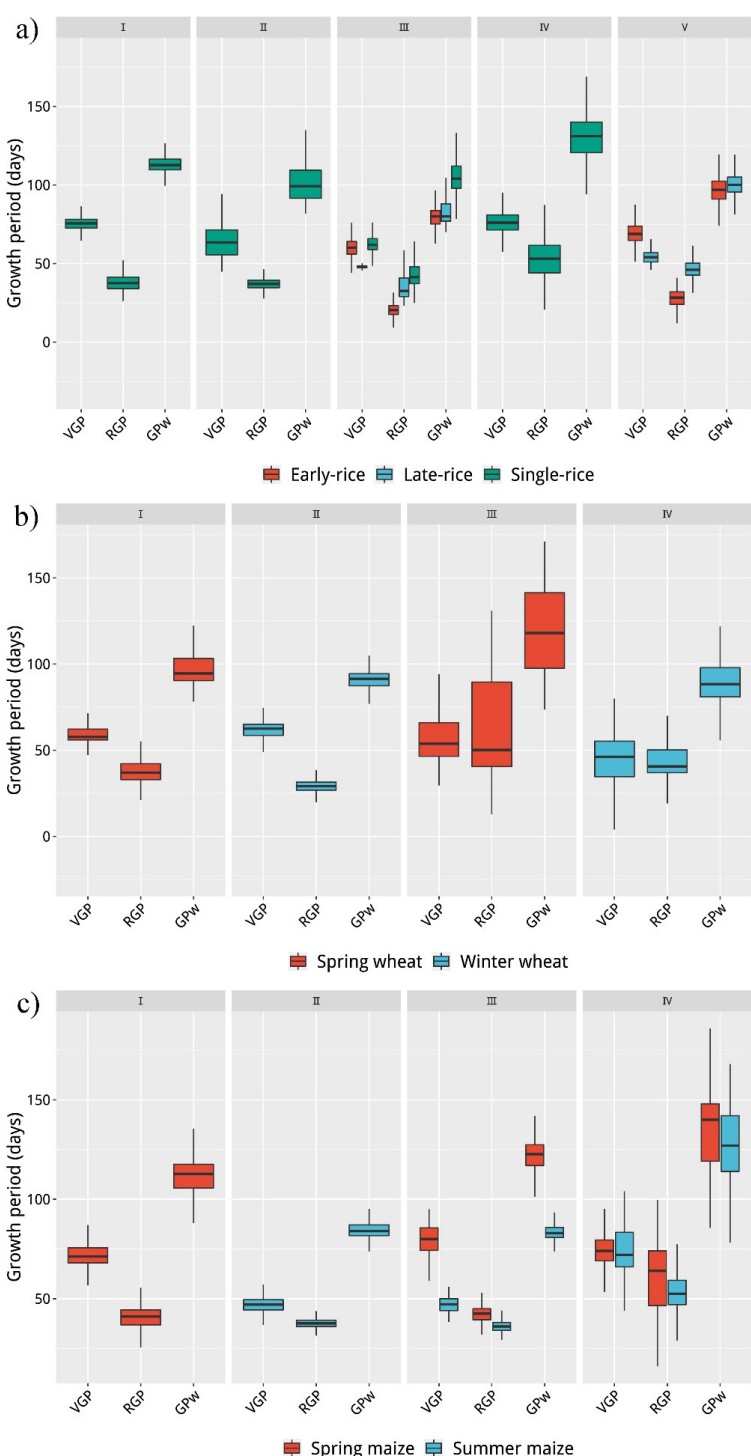

**Fig.8:** Box plots of three key phenological periods by crop and sub-regions during 2000~2015, for vegetative growth period (VGP), reproductive growth period (VGP), and whole growth period (GPW) of rice (a), wheat (b), and maize (c).

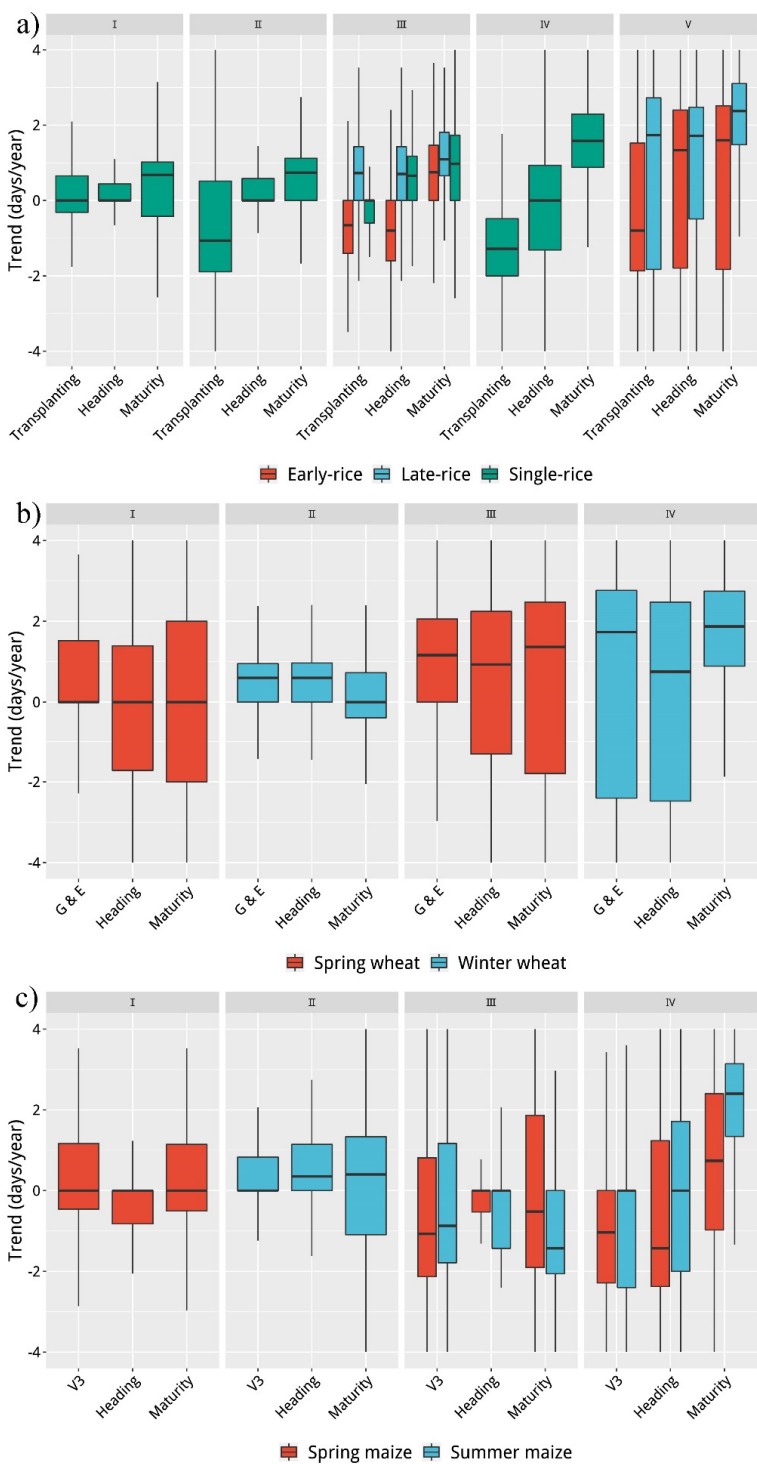

**Fig.9: The trends of three key phenological dates during 2000~2015 by crop and sub-regions during, VGP for vegetative growth period, RGP for reproductive growth period, and GPW for whole growth period.**

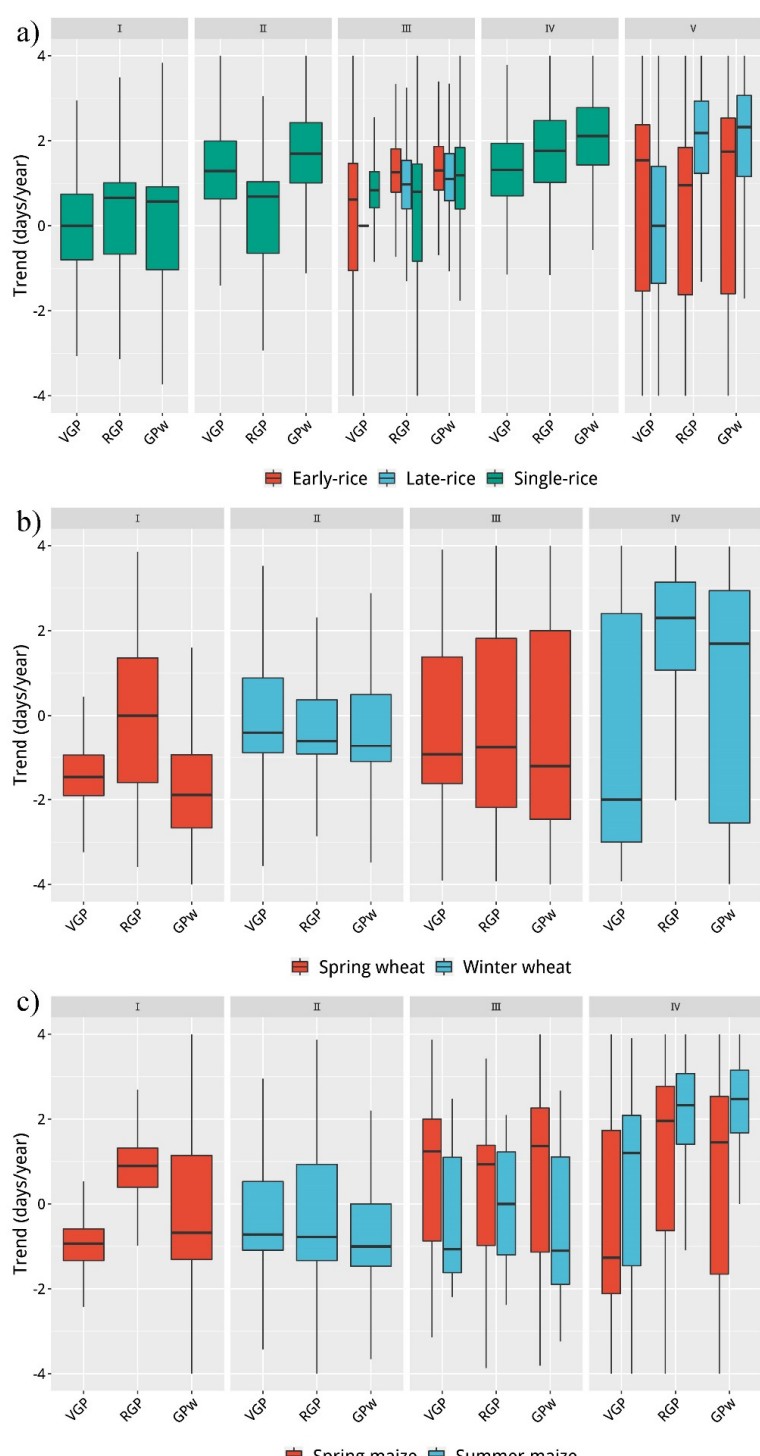


**Fig.10: The trends of three key phenological periods during 2000~2015 by crop and sub-regions during, VGP for vegetative growth period, RGP for reproductive growth period, and GPW for whole growth period.**