# Peer review of "ChinaCropPhen1km: A high-resolution crop phenological dataset for three staple crops in China during 2000-2015 based on LAI products"

_Earth System Science Data, 2019_

## Referee Comment (RC1) · Anonymous Referee #1 · 10 Sep 2019

Accurate crop phenological dataset at the large regional scale is of great importance to various agricultural applications. This paper established the ChinaCropPhen1km, which consists of high-resolution gridded-phenology product for three major staple crops of China, i.e., wheat, maize, rice. Several comments for this paper are listed as follows.

1) As the authors pointed out, the study area of this study (i.e., the China mainland) possesses complex environments and crop planting patterns, diverse cropping intensity and cultivation habits. Therefore, according to previous studies, I suggest the practice of separating the whole study area into sub agro-climatic zones, and estab-

[Figure]

lishing model for each crop type from each zone, might further improve the model's performance and dataset accuracy.

2) Another issue is that how the authors determine the spatial distribution of each crop type. Any crop mask have been used in your study?

3) Besides, the uncertainty of the developed dataset should be further discussed. As the proposed dataset are based on the GLASS LAI, it is suggested that the accuracy of GLASS LAI should also be provided. And the authors had better analyze the impact of the uncertainties of GLASS LAI on ChinaCropPhen1km.

---

## Referee Comment (RC2) · Anonymous Referee #2 · 14 Oct 2019

General Comments: This study provides a long-term phenology dataset for three staple crops in China at the 1km spatial resolution (ChinaCropPhen1km). Such dataset with high resolution and accuracy is very useful to the researchers focused on crop model, yield estimation, food security, impact evaluation from climate change, and etc.. Meanwhile, the method proposed is robust and repeatable, and the authors' study provide a potential tool to apply into other regions and other crop systems. The manuscript is generally well-structured and well-written, and many findings are very interesting and very attractive to many potential readers. The manuscript falls well within the scope of the journal and provides a suitable contribution to ESSD. Therefore, I recommend it can be acceptable for publication with minor revision. Âǎ The following specific com-

ments should be noticed: 1)ÂăÂăÂăÂă Line 39: "estimate" should be "estimated". 2)ÂăÂăÂăÂă Line 53: Add "the" before "essential". 3)ÂăÂăÂăÂă Line 69: "the single cropping system of spring maize in Northeast China" is not an example of "multi-cropped". 4)ÂăÂăÂăÂă Line 91: Replace "including" with "i.e.,". 5)ÂăÂăÂăÂă Line 95: Replace "smooth" with "smoothing". 6)ÂăÂăÂăÂă Line 169: Singularize "crop pixels". 7)ÂăÂăÂăÂă Line 170: Add "using" before "RMSE". 8)ÂăÂăÂăÂă Line 189: Revise "the median columns of Fig.3" to detailed icon of specifiedÂăfigure. 9)ÂăÂăÂăÂă Line 196: In terms of "the uncertainty of GLASS LAI data", how such uncertainties may affect the generated dataset? 10)Âă Line 212: Delete "degrees". 11)Âă Line 217: Add "be" before "ignored". 12)Âă Line 274: V3 is not the key phenological stage for wheat, delete "V3" in parenthesis. 13)Âă Line 282: Add "for" before "wheat"

---

## Author Comment (AC1) · 13 Nov 2019

Thanks for your careful reviewing and all constructive comments on our manuscript. We have taken all your comments into account and responded positively to qualify our manuscript for a potential publication in the journal. The reviewer's comments are marked in black color, while our detailed responses in blue. Accurate crop phenological dataset at the large regional scale is of great importance to various agricultural applications. This paper established the ChinaCropPhen1km, which consists of high-resolution gridded-phenology product for three major staple crops of China, i.e., wheat, maize, rice. Several comments for this paper are listed as follows. 1) As the authors

pointed out, the study area of this study (i.e., the China mainland) possesses complex environments and crop planting patterns, diverse cropping intensity and cultivation habits. Therefore, according to previous studies, I suggest the practice of separating the whole study area into sub agro-climatic zones, and establishing model for each crop type from each zone, might further improve the model's performance and dataset accuracy. Response: Thank you a lot for the insightful suggestion. We greatly agree with you that the studied area division is vital for detecting accurately crop phenology owning to the complex environments and crop planting patterns, diverse cropping intensity and cultivation habits in the study area. Actually, we separated the whole studied area into different provincial administrative units, which is more specific than agricultural zones. Especially, for provinces with large heterogeneity in crop planting patterns, such as Shaanxi, Shanxi and Hebei province where spring-sown crops are planted in the north and summer-sown crops in the south due to the different agroclimatic characteristics, we divided them into two parts based on agroclimatic regionalization to detect phenology for each crop type (as shown in Fig.1). Given the large spatial extent of the study area, we believe that the method of studied area division used in our study is more specific and reasonable than what you have suggested. Reference: Zhao, J., Yang, X., & Sun, S. (2018). Constraints on maize yield and yield stability in the main cropping regions in China. European Journal of Agronomy, 106-115. 2) Another issue is that how the authors determine the spatial distribution of each crop type. Any crop mask have been used in your study? Response: Yes, we did use crop mask for identifying the spatial distribution of each crop type. In this study, we first selected the cultivated-land layer derived from the 1-km National Land Cover Dataset (NLCD) of China as cropland masks. Then, we identified the inflection and maximum points of LAI time-series for each cropland grid as indicators of corresponding key phenological stages (as mentioned in Section 2.3.3) for each crop within the restricted time windows based on the observations around the nearest AMS. Finally, we regarded the grids with three indicators during the time windows as crop-cultivated grids for each crop. Specifically, we detected the key phenological dates for dryland crops (i.e., maize and wheat) and

paddy rice, which were restricted on the dry land and paddy field layer derived from the NLCD, respectively. Reference: Chen, Y., Zhang, Z., & Tao, F. (2018). Improving regional winter wheat yield estimation through assimilation of phenology and leaf area index from remote sensing data. European Journal of Agronomy, 163-173.

Fig.1: Spatial patterns of annual averages of three key phenological dates during 2000∼2015 for rice (a), wheat (b), and maize (c). 3) Besides, the uncertainty of the developed dataset should be further discussed. As the proposed dataset are based on the GLASS LAI, it is suggested that the accuracy of GLASS LAI should also be provided. And the authors had better analyze the impact of the uncertainties of GLASS LAI on ChinaCropPhen1km. Response: Thanks very much for your constructive comment. We have followed you to insert relevant contents into our manuscript (highlighted in "Track Changes" as suggested in the revised manuscript). 1. In the Data and methods (Section 2.2.1), the accuracy of GLASS LAI has been provided from Line 78 to 80 in the revised manuscript. 2. In the Results and Discussion (Section 3.5), we added one paragraph (from Line 296 to 316 in the revised manuscript) for discussing the uncertainties in the study.

Please also note the supplement to this comment:
https://www.earth-syst-sci-data-discuss.net/essd-2019-110/essd-2019-110-AC1-supplement.pdf
* * *
N

Hebei

Shanxi

Shaanxi

☐ Maize Subregion Boundary
☐ Provincial Administrative Boudary

0  195 390      780      1,170    1,560
                                        Miles

**Fig. 1.**

[Figure]

**Supplement:**

Thanks for your careful reviewing and all constructive comments on our manuscript. We have taken all your comments into account and responded positively to qualify our manuscript for a potential publication in the journal. The reviewer's comments are marked in black color, while our detailed responses in blue. Accurate crop phenological dataset at the large regional scale is of great importance to various agricultural applications. This paper established the ChinaCropPhen1km, which consists of high-resolution gridded-phenology product for three major staple crops of China, i.e., wheat, maize, rice. Several comments for this paper are listed as follows.

1) As the authors pointed out, the study area of this study (i.e., the China mainland) possesses complex environments and crop planting patterns, diverse cropping intensity and cultivation habits. Therefore, according to previous studies, I suggest the practice of separating the whole study area into sub agro-climatic zones, and establishing model for each crop type from each zone, might further improve the model's performance and dataset accuracy.

**Response:** Thank you a lot for the insightful suggestion. We greatly agree with you that the studied area division is vital for detecting accurately crop phenology owning to the complex environments and crop planting patterns, diverse cropping intensity and cultivation habits in the study area. Actually, we separated the whole studied area into different provincial administrative units, which is more specific than agricultural zones. Especially, for provinces with large heterogeneity in crop planting patterns, such as Shaanxi, Shanxi and Hebei province where spring-sown crops are planted in the north and summer-sown crops in the south due to the different agroclimatic characteristics, we divided them into two parts based on agroclimatic regionalization to detect phenology for each crop type (as shown in Fig.1). Given the large spatial extent of the study area, we believe that the method of studied area division used in our study is more specific and reasonable than what you have suggested.

**Reference:**

Zhao, J., Yang, X., & Sun, S. (2018). Constraints on maize yield and yield stability in the main cropping regions in China. European Journal of Agronomy, 106-115.

[Figure]

Fig.1: Spatial patterns of annual averages of three key phenological dates during 2000~2015 for rice (a), wheat (b), and maize (c).

2) Another issue is that how the authors determine the spatial distribution of each crop type. Any crop mask have been used in your study?

**Response:** Yes, we did use crop mask for identifying the spatial distribution of each crop type. In this study, we first selected the cultivated-land layer derived from the 1-km National Land Cover Dataset (NLCD) of China as cropland masks. Then, we identified the inflection and maximum points of LAI time-series for each cropland grid as indicators of corresponding key phenological stages (as mentioned in Section 2.3.3) for each crop within the restricted time windows based on the observations around the nearest AMS. Finally, we regarded the grids with three indicators during the time windows as cropcultivated grids for each crop. Specifically, we detected the key phenological dates for dryland crops (i.e., maize and wheat) and paddy rice, which were restricted on the dry land and paddy field layer derived from the NLCD, respectively.

[revised manuscript text omitted]

---

## Author Comment (AC2) · 13 Nov 2019

Thanks for your careful review and constructive comments on our manuscript, and they really inspire us to improve the paper's quality. We have taken all your comments into account and responded positively to qualify our manuscript for a potential publication in the journal. The reviewer's comments are marked in black, while our detailed responses in blue. General Comments: This study provides a long-term phenology dataset for three staple crops in China at the 1km spatial resolution (ChinaCrop-Phen1km). Such dataset with high resolution and accuracy is very useful to the researchers focused on crop model, yield estimation, food security, impact evaluation

from climate change, and etc.. Meanwhile, the method proposed is robust and repeatable, and the authors' study provide a potential tool to apply into other regions and other crop systems. The manuscript is generally well-structured and well-written, and many findings are very interesting and very attractive to many potential readers. The manuscript falls well within the scope of the journal and provides a suitable contribution to ESSD. Therefore, I recommend it can be acceptable for publication with minor revision. The following specific comments should be noticed: 1) Line 39: "estimate" should be "estimated". Response: Many thanks for your careful check and valuable comment. We have followed your advice and it will be modified in the revised paper. 2) Line 53: Add "the" before "essential". Response: Thanks for your careful review. It will be modified in the revised paper (Line 54). 3) Line 69: "the single cropping system of spring maize in Northeast China" is not an example of "multi-cropped". Response: Thanks for your careful review. We have deleted "the single cropping system of spring maize in Northeast China" in Line 70. 4) Line 91: Replace "including" with "i.e.,". Response: Thanks for your careful review. It will be modified in the revised paper (Line 95). 5) Line 95: Replace "smooth" with "smoothing". Response: Many thanks for your careful check. We have corrected this mistake in the new edition. (Line 99). 6) Line 169: Singularize "crop pixels". Response: Thank you so much for your comments. We have followed your advice and revised it in the revised paper (Line 173). 7) Line 170: Add "using" before "RMSE". Response: Thank you for your comments. It will be modified in the revised paper (Line 174). 8) Line 189: Revise "the median columns of Fig.3" to detailed icon of specified figure Response: Thank you for your careful comments. We have specified the corresponding icon of figure (i.e., Fig. 3-b, f) in the revised paper (Line 193). 9) Line 196: In terms of "the uncertainty of GLASS LAI data", how such uncertainties may affect the generated dataset? Response: Thank you a lot for the insightful suggestion. We have followed your suggestion and added relevant contents into our manuscript (highlighted in "Track Changes" as suggested in the revised manuscript). In the Results and Discussion (Section 3.5), we added one paragraph (from Line 296 to 316 in the revised manuscript) for discussing the

uncertainties in the study. 10) Line 212: Delete "degrees". Response: Thank you for your careful comments. It will be modified in the revised paper (Line 216). 11) Line 217: Add "be" before "ignored". Response: Many thanks for your careful check. We have corrected this mistake in the new edition. (Line 221). 12) Line 274: V3 is not the key phenological stage for wheat, delete "V3" in parenthesis. Response: Thanks a lot for your careful comments. It will be modified in the revised paper (Line 278). 13) Line 282: Add "for" before "wheat". Response: Thanks for your careful check. We have followed your advice and revised it in the revised paper (Line 286).

Please also note the supplement to this comment:
https://www.earth-syst-sci-data-discuss.net/essd-2019-110/essd-2019-110-AC2-supplement.pdf

[Figure]

**Supplement:**

Thanks for your careful review and constructive comments on our manuscript, and they really inspire us to improve the paper's quality. We have taken all your comments into account and responded positively to qualify our manuscript for a potential publication in the journal. The reviewer's comments are marked in black, while our detailed responses in blue.

General Comments:

This study provides a long-term phenology dataset for three staple crops in China at the 1km spatial resolution (ChinaCropPhen1km). Such dataset with high resolution and accuracy is very useful to the researchers focused on crop model, yield estimation, food security, impact evaluation from climate change, and etc.. Meanwhile, the method proposed is robust and repeatable, and the authors' study provide a potential tool to apply into other regions and other crop systems. The manuscript is generally well-structured and well-written, and many findings are very interesting and very attractive to many potential readers. The manuscript falls well within the scope of the journal and provides a suitable contribution to ESSD. Therefore, I recommend it can be acceptable for publication with minor revision.

The following specific comments should be noticed:

1) Line 39: "estimate" should be "estimated".

    **Response:** Many thanks for your careful check and valuable comment. We have followed your advice and it will be modified in the revised paper.

2) Line 53: Add "the" before "essential".

    **Response:** Thanks for your careful review. It will be modified in the revised paper (Line 54).

3) Line 69: "the single cropping system of spring maize in Northeast China" is not an example of "multi-cropped".

    **Response:** Thanks for your careful review. We have deleted "the single cropping system of spring maize in Northeast China" in Line 70.

4) Line 91: Replace "including" with "i.e.,".

    **Response:** Thanks for your careful review. It will be modified in the revised paper (Line 95).

5) Line 95: Replace "smooth" with "smoothing".

    **Response:** Many thanks for your careful check. We have corrected this mistake in the new edition. (Line 99).

6) Line 169: Singularize "crop pixels".

   **Response:** Thank you so much for your comments. We have followed your advice and revised it in the revised paper (Line 173).

7) Line 170: Add "using" before "RMSE".

   **Response:** Thank you for your comments. It will be modified in the revised paper (Line 174).

8) Line 189: Revise "the median columns of Fig.3" to detailed icon of specified figure

   **Response:** Thank you for your careful comments. We have specified the corresponding icon of figure (i.e., Fig. 3-b, f) in the revised paper (Line 193).

9) Line 196: In terms of "the uncertainty of GLASS LAI data", how such uncertainties may affect the generated dataset?

   **Response:** Thank you a lot for the insightful suggestion. We have followed your suggestion and added relevant contents into our manuscript (highlighted in "Track Changes" as suggested in the revised manuscript). In the Results and Discussion (Section 3.5), we added one paragraph (from Line 296 to 316 in the revised manuscript) for discussing the uncertainties in the study.

10) Line 212: Delete "degrees".

    **Response:** Thank you for your careful comments. It will be modified in the revised paper (Line 216).

11) Line 217: Add "be" before "ignored".

    **Response:** Many thanks for your careful check. We have corrected this mistake in the new edition. (Line 221).

12) Line 274: V3 is not the key phenological stage for wheat, delete "V3" in parenthesis.

    **Response:** Thanks a lot for your careful comments. It will be modified in the revised paper (Line 278).

13) Line 282: Add "for" before "wheat".

    **Response:** Thanks for your careful check. We have followed your advice and revised it in the revised paper (Line 286).

[revised manuscript text omitted]

---

## Author Response (AR3)

Dear editor and reviewers:

Thank you for the careful handling of the manuscript and all constructive comments that substantially increased the quality of our manuscript. Based on the critical comments and thoughtful suggestion of reviewers, we have made carefully revisions and include all changes in the revised version of the manuscript. Additionally, we have modified the section "Acknowledgements".

Best regards,

Zhao Zhang

**Response to Topical Editor:**

Thanks for your careful reviewing and all constructive comments on our manuscript. We have taken all your comments into account and responded positively to qualify our manuscript for a potential publication in the journal. The editor's comments are marked in black color, while our detailed responses in blue.

Formal Requirements

In general

I) please spell out all abbreviations, when abbreviations appear i) in the abstract, e.g. L11 Global Land Surface Satellite (GLASS) ii) when they first appear in the manuscript text, e.g., RMSE, CYCLOPES, Ensemble of Satellites provide information on it ánd links and or references, same for HJ-1 A/B CCD, GEOV1

**Response:** Thank you a lot for the careful check. We have spelt out all abbreviations in the revised manuscript.

1) Line 14: we replace "GLASS LAI" by "Global Land Surface Satellite (GLASS) Leaf Area Index (LAI)";

2) Line 37: we replace "HJ-1 A/B CCD" by "Environment and Disaster Monitoring and Prediction

Satellite Constellation A/B (HJ-1 A/B) charge coupled device (CCD) sensor";

3) Line 84-85: we replace "CYCLOPES" by "Carbon cYcle and Change in Land Observational Products from an Ensemble of Satellites (CYCLOPES)";

3) Line 85-86: we replace "BELMANIP" by "BEnchmark Land Multisite ANalysis and Intercomparison of Products (BELMANIP)";

4) Line 87: we replace "RMSE" by "Root Mean Square Error (RMSE)"; replace "$R^2$" by "determination coefficients ($R^2$)";

5) Line 89: we replace "MOD15" by "MODIS LAI product (MOD15)"; replace "GEOV1" by "Geoland2/BioPar version 1 (GEOV1)".

II) please use formal writing L30 can't -> cannot, LL236 wasn't, L261 didn't …

**Response:** Thank you a lot for the careful check. We have modified them in the revised paper.

1) Line 32: can't -> cannot
2) Line 199: it's -> it is
3) Line 256: wasn't -> was not
4) Line 282: didn't -> did not

Please consider: As Earth System Science Data (ESSD) is the journal for the publication on data products your manuscript would require in some chapters, specifically in abstract and conclusions and in some titles more the focus on the data set then on a study describing a method

e.g., in Abstract and conclusions

'In this study, we proposed a method to retrieve 1km-grid crop phenological dataset for three main crops from 2000 to 2015 based on GLASS LAI products…

…Finally, we established a high resolution gridded-phenology product for three staple crops in China during 2000-2015, named as ChinaCropPhen1km'

Please change accordingly starting with your data product and the focus on the data product 'In this study, we produced a high resolution gridded-phenology product for three staple crops in China during 2000-2015 based on GLASS MODIS LAI products…, named as ChinaCropPhen1km' and similar

**Response:** Thanks very much for your constructive comment. We have followed your advice and modified it in the revised paper.

1) Line 13-15: we have modified the sentence to "In this study, we produced a 1km-grid crop phenological dataset for three main crops from 2000 to 2015 based on Global Land Surface Satellite (GLASS) Leaf Area Index (LAI) products, named as ChinaCropPhen1km.", and deleted ", named as ChinaCropPhen1km" in Line 19.

2) Line 221: we have changed the title of section 3.2 "Validation of the phenological data" to "Validation of ChinaCropPhen1km"

3) Line 335-336: We have modified the sentence to "In the present study, we generated 1km-grid crop phenological dataset for three main crops from 2000 to 2015 based on GLASS LAI products, named as ChinaCropPhen1km.", and deleted ", i.e., ChinaCropPhen1km" in Line 341.

Introduction

L46 It is urgently required to acquire the gridded phenological datasets -> remains unclear – please reword the sentence

**Response:** Thanks very much for your careful check. We have reworded the sentence to "To implement a large-scale agricultural system simulation for multiple crops, there is an urgent need to acquire the gridded phenological dataset for each crop at a national or global scale." (Line 49-50).

Chapter 2 Data and Methods

L66 The study areas are across the mainland of China, possessing of complex environments
please reword the sentence, e.g. a.. areas .. are characterised by different …

**Response:** Thanks very much for your careful check. We have reworded the sentence to "The study areas across the mainland of China are characterized by complex environments and crop planting structures, diverse cropping intensity and cultivation habits." (Line 70-71).

Here is also the right place to introduce the subregions used later in results and discussions

Please include a figure also in manuscript text text showing the provinces I, II, III, IV, V -now in supplement S1 only, figure similar as in supplement S1, please edit the unit of the scale bar: change miles to km

**Response:** Thank you a lot for the insightful suggestion. We have added two sentences to introduce the subregions (Line 73-74) and Table S1-S3 in Supplement to provide the details of the subregions for each crop. Meanwhile, we have changed the Fig. 1 to show the divided subregions for each crop. Finally, we have edited the unit of the scale bar for Fig. 6 and Fig. S1-S4.

2.2.1 change title to ChinaCropPhen Input data

Put in here The GLASS LAI satellite product data used as input for phenology

An improved MODIS-based LAI dataset (GLASS LAI) from 2000 to 2015 was from Liang et al. (2013)

And we suggest also to put in this subchapter the National Land Cover Dataset (NLCD) that is used as a mask for 'labelling' / assigning the crop type as it belongs to the production of the data product

Can you go in detail in methods how NLCD is used for the masking

**Response:** Thanks very much for your constructive comment. We have followed your advice and modified it in the revised paper (Line 81-97). We added the sentence "In addition, the cultivated-land layer derived from the 1-km National Land Cover Dataset (NLCD) of China was used as cropland masks. Specifically, we detected the key phenological dates for dryland crops (i.e., maize and wheat) and paddy rice, which were restricted on the dry land and paddy field layer derived from the NLCD, respectively." to provide the detail in methods how NLCD is used for masking.

2.2.2 ChinaCrop Phen validation data

**Response:** Thanks very much for your insightful comment. We have changed the title as your advice suggests.

2.3. Methods chosen to smooth LAI products -> ChinaCropPhen LAI smoothing methods

Three popular methods -> change to three commonly used methods

**Response:** Thanks very much for your constructive suggestion. We have followed your advice and

modified it in the revised paper (Line 120-124).

3.4 The changes in three key phenological dates and growth periods from 2000 to 2015

2.3.5 Retrieving the phenological information at 1-km pixel across China -> ChinaCropPhen
L168 … OFP approach and regarded the grids that the three key phenological stages (mentioned in 2.3.3) could be simultaneously … - remains unclear, please rephrase

**Response:** Thanks very much for your constructive suggestion and careful check. We have changed the title of section 2.3.5 to "Generating ChinaCropPhen1km dataset". Moreover, we have modified the sentence and added an example to explain clearly (Line 185-189).

chapter 3.5 discussion on uncertainty
please check the English throughout this chapter, e.g.

**Response:** Thanks a lot for your careful check. We have reorganized the structure of this chapter carefully (Line 315-329). We summarize the sources of uncertainties in ChinaCropPhen1km to two aspects. One is the quality of GLASS LAI products (Line 315-322). The other is the mixed pixel issue (Line 323-329). The former can be divided into two secondary aspects, i.e., the noise of original GLASS LAI time-series (Line 316-319) and deficiency in GLASS LAI retrieval algorithm (Line 320-322). Similarly, the later was resulted from the coarse spatial resolution of 1 km (Line 325-326) and the inclusion of several crop types in the dryland layer of NLCD (Line 326-327).

First of all, the uncertainties of the GLASS LAI products have a relatively greater impact on hinaCropPhen1km – please rephrase wording – greater impact than what?

**Response:** Thanks a lot for your careful check. We have reworded this sentence to "On the one hand, GLASS LAI products might lead to some uncertainties in ChinaCropPhen1km.".

the unavailable crop-specific map ? not clear

**Response:** Thanks a lot for your careful check. We have changed the phrase to "the crop-specific map".

conclusions

In conclusion please include more characteristics and results on the data set,

1 km , evaluated –better than 10 days

**Response:** Thanks very much for your constructive suggestion. We have added the evaluated accuracy of ChinaCropPhen1km in Line 343.

Figures

figure 2 this work flow is a good idea and intuitive and well understandable by the readers. However, you can optimize the figure
suggestions
• land use products -> you could name it NLCD
• can you indicate how NLCD is used for the masking, e,g by placing a box with 'mask' on the line leading to NLCD
• please rephrase in a more technical way 'the most suitable smoothing method'

in the figure caption provide description of short names

**Response:** Thanks very much for your constructive suggestion. We have followed your advice and optimized the Figure 2.

Fig 4
Exchange raw data with

**Response:** Thanks very much for your careful check. We have replaced the "raw data" in Fig. 4 by "Original GLASS LAI".

Figure 5 x – axis add recorded AMS dates

**Response:** Thanks very much for your constructive suggestion. We have followed your advice and modified the title of x-axis.

Table 1: heading of left column region ? or province/subprovince?

**Response:** Thanks very much for your careful check. We have added the "province" to the heading of left column.

Data publication on figshare
ChinaCropPhen1km geotiffs are well readable and contain good meta data

Please provide the information on the projection and format already in the figshare abstract
Description that the pixel value is the Julian Day of the Year DOY
You can also consider to change the zero value to a No data value.
**Response:** Thanks very much for your careful check and constructive suggestion. We have updated the abstract of figshare and the dataset as your advice suggests.

**Response to Reviewer #1:**

Thanks for your careful reviewing and all constructive comments on our manuscript. We have taken all your comments into account and responded positively to qualify our manuscript for a potential publication in the journal. The reviewer's comments are marked in black color, while our detailed responses in blue. Accurate crop phenological dataset at the large regional scale is of great importance to various agricultural applications. This paper established the ChinaCropPhen1km, which consists of high-resolution gridded-phenology product for three major staple crops of China, i.e., wheat, maize, rice. Several comments for this paper are listed as follows.

1) As the authors pointed out, the study area of this study (i.e., the China mainland) possesses complex environments and crop planting patterns, diverse cropping intensity and cultivation habits. Therefore, according to previous studies, I suggest the practice of separating the whole study area into sub agro-climatic zones, and establishing model for each crop type from each zone, might further improve the model's performance and dataset accuracy.

**Response:** Thank you a lot for the insightful suggestion. We greatly agree with you that the studied area division is vital for detecting accurately crop phenology owning to the complex environments and crop planting patterns, diverse cropping intensity and cultivation habits in the study area. Actually, we separated the whole studied area into different provincial administrative units, which is more specific than

agricultural zones. Especially, for provinces with large heterogeneity in crop planting patterns, such as Shaanxi, Shanxi and Hebei province where spring-sown crops are planted in the north and summer-sown crops in the south due to the different agroclimatic characteristics, we divided them into two parts based on agroclimatic regionalization to detect phenology for each crop type (as shown in Fig.1). Given the large spatial extent of the study area, we believe that the method of studied area division used in our study is more specific and reasonable than what you have suggested.

**Reference:**

Zhao, J., Yang, X., & Sun, S. (2018). Constraints on maize yield and yield stability in the main cropping regions in China. European Journal of Agronomy, 106-115.

[Figure]

Fig.1: Spatial patterns of annual averages of three key phenological dates during 2000~2015 for rice (a),

wheat (b), and maize (c).

2) Another issue is that how the authors determine the spatial distribution of each crop type. Any crop mask have been used in your study?

**Response:** Yes, we did use crop mask for identifying the spatial distribution of each crop type. In this study, we first selected the cultivated-land layer derived from the 1-km National Land Cover Dataset (NLCD) of China as cropland masks. Then, we identified the inflection and maximum points of LAI time-series for each cropland grid as indicators of corresponding key phenological stages (as mentioned in Section 2.3.3) for each crop within the restricted time windows based on the observations around the nearest AMS. Finally, we regarded the grids with three indicators during the time windows as crop-cultivated grids for each crop. Specifically, we detected the key phenological dates for dryland crops (i.e., maize and wheat) and paddy rice, which were restricted on the dry land and paddy field layer derived from the NLCD, respectively.

**Response:** Thanks for your careful check. We have followed your advice and revised it in the revised paper (Line 304).

[revised manuscript text omitted]